# Liuwei Dihuang Pills Enhance Osteogenic Differentiation in MC3T3-E1 Cells through the Activation of the Wnt/β-Catenin Signaling Pathway

**DOI:** 10.3390/ph17010099

**Published:** 2024-01-11

**Authors:** Jinlong Zhao, Guihong Liang, Junzheng Yang, Hetao Huang, Yaoxing Dou, Zhuoxu Gu, Jun Liu, Lingfeng Zeng, Weiyi Yang

**Affiliations:** 1State Key Laboratory of Traditional Chinese Medicine Syndrome, The Second Clinical College of Guangzhou University of Chinese Medicine, Guangzhou 510405, China; 20222110142@stu.gzucm.edu.cn (J.Z.); liangguihong@gzucm.edu.cn (G.L.); yaoxingdou@126.com (Y.D.); 20231110746@stu.gzucm.edu.cn (Z.G.); 2Guangdong Provincial Key Laboratory of Chinese Medicine for Prevention and Treatment of Refractory Chronic Diseases, The Second Affiliated Hospital of Guangzhou University of Chinese Medicine (Guangdong Provincial Hospital of Chinese Medicine), Guangzhou 510120, China; ynhtsmile@126.com; 3The Research Team on Bone and Joint Degeneration and Injury, Guangdong Provincial Academy of Chinese Medical Sciences, Guangzhou 510120, China; gjtb@gzucm.edu.cn; 4The Fifth Clinical College, Guangzhou University of Chinese Medicine, Guangzhou 510405, China; 20201110029@stu.gzucm.edu.cn; 5Guangdong Second Chinese Medicine Hospital (Guangdong Province Engineering Technology Research Institute of Traditional Chinese Medicine), Guangzhou 510095, China

**Keywords:** Liuwei Dihuang pills, osteogenic, osteoporosis, MC3T3-E1 cells, Wnt/β-catenin signaling pathway

## Abstract

Objective: The therapeutic efficacy and molecular mechanisms of traditional Chinese medicines (TCMs), such as Liuwei Dihuang pills (LWDH pills), in treating osteoporosis (OP) remain an area of active research and interest in modern medicine. This study investigated the mechanistic underpinnings of LWDH pills in the treatment of OP based on network pharmacology, bioinformatics, and in vitro experiments. Methods: The active ingredients and targets of LWDH pills were retrieved through the TCMSP database. OP-related targets were identified using the CTD, GeneCards, and DisGeNET databases. The STRING platform was employed to construct a protein–protein interaction (PPI) network, and core targets for LWDH pills in treating OP were identified. The GO functional and KEGG pathway enrichment analyses for potential targets were performed using the R package “clusterProfiler”. A “drug–target” network diagram was created using Cytoscape 3.7.1 software. The viability of MC3T3-E1 cells was evaluated using the CCK-8 method after treatment with various concentrations (1.25%, 2.5%, 5%, and 10%) of LWDH pill-medicated serum for 24, 48, and 72 h. Following a 48 h treatment of MC3T3-E1 cells with LWDH pill-medicated serum, the protein levels of collagen Ⅰ, RUNX2, Wnt3, and β-catenin were quantified using the Western blot analysis, and the activity of alkaline phosphatase (ALP) was measured. Results: A total of 197 putative targets for LWDH pills for OP treatment were pinpointed, from which 20 core targets were singled out, including *TP53*, *JUN*, *TNF*, *CTNNB1* (β-catenin), and *GSK3B*. The putative targets were predominantly involved in signaling pathways such as the Wnt signaling pathway, the MAPK signaling pathway, and the PI3K-Akt signaling pathway. The intervention with LWDH pill-medicated serum for 24, 48, and 72 h did not result in any notable alterations in the cell viability of MC3T3-E1 cells relative to the control group (all *p* > 0.05). Significant upregulation in protein levels of collagen Ⅰ, RUNX2, Wnt3, and β-catenin in MC3T3-E1 cells was observed in response to the treatment with 2.5%, 5%, and 10% of LWDH pill-medicated serum in comparison to that with the 10% rabbit serum group (all *p <* 0.05). Furthermore, the intervention with LWDH pill-medicated serum resulted in the formation of red calcified nodules in MC3T3-E1 cells, as indicated by ARS staining. Conclusions: LWDH pills may upregulate the Wnt/β-catenin signaling pathway to elevate the expression of osteogenic differentiation proteins, including collagen Ⅰ and RUNX2, and to increase the ALP activity in MC3T3-E1 cells for the treatment of OP.

## 1. Introduction

Osteoporosis (OP) is a common systemic metabolic disorder affecting bone health, characterized by a decrease in bone density, loss of bone mass, and deterioration of the bone microstructure. These factors collectively result in diminished bone strength and an elevated susceptibility to fragile bone fractures [1,2]. The most serious complication of OP is osteoporotic fracture, which primarily occurs in the thoracolumbar vertebrae, hip, and distal radius [3,4]. The mortality rate of patients with OP may reach 15–20% within one year following a hip fracture, and this elevated risk of death persists for at least five years [3,4]. Based on the diagnostic criteria established by the World Health Organization (WHO), recent epidemiological studies have revealed a global prevalence rate of 19.7% for OP [5,6]. Moreover, diverse prevalence rates of OP exist across different nations (4.1% in the Netherlands to 52.0% in Turkey) and continents (8.0% in Oceania to 26.9% in Africa) [5,6]. Undoubtedly, the prevention and treatment of OP is a public health challenge faced by the entire human population. At present, commonly used anti-OP medications in clinical settings have various limitations or adverse reactions, including the need for long-term administration, osteonecrosis of the jaw, and gastrointestinal adverse reactions [7,8,9]. Consequently, the continuous exploration for more effective and safer anti-OP treatments remains a valuable endeavor.

Traditional Chinese medicine (TCM) is extensively used in China to treat various ailments, including OP. Liuwei Dihuang pills (LWDH pills) are a formulated medicine composed of six herbal ingredients: prepared Rehmannia root, *Cornus officinalis*, *Cortex Moutan*, *Rhizoma Dioscoreae*, *Poria cocos*, and *Alismatis Rhizoma* [10]. These herbal ingredients have been extensively recognized to confer therapeutic properties in TCM that include nourishing yin, tonifying the kidneys, replenishing the essence, and benefiting the marrow; in other words, they can promote the balance and stability of the human body’s internal environment and enhance the activity of osteoblasts. According to TCM theory, OP is primarily attributed to kidney deficiency, which causes insufficient nourishment of the bones and ultimately leads to the development of OP. The efficacy of LWDH pills in the treatment of OP has been established based on existing evidence [11,12]. Nevertheless, the complex composition of LWDH pills has posed challenges in fully understanding their pharmacological mechanism. Research indicates the involvement of various transcription factors and cytokines, including the Wnt/β-catenin, TGF/BMP, and MAPK signaling pathways, in the regulation of bone remodeling [13,14]. Among these pathways, the Wnt/β-catenin signaling pathway is considered the most canonical. Research has documented that the activation of the Wnt/β-catenin signaling pathway in the human body can initiate the process of osteoblast differentiation. This activation effectively amplifies the functionality and mineralization capability of osteoblasts, thereby resulting in a notable augmentation of bone mass and an improved ability to endure mechanical stress on the bone. Moreover, the Wnt signaling pathway can induce osteogenesis by upregulating RUNX2. Furthermore, the overexpression of Wnt10b has the capacity to curtail bone loss and drive mesenchymal stem cell (MSC) differentiation by upregulating RUNX2 [15,16]. Therefore, exploring the impact of the Wnt/β-catenin signaling pathway on the process of osteogenic differentiation offers promising insights into the pharmacological mechanism of LWDH pills.

In this study, our investigation involves an integrated application of network pharmacology, bioinformatics, and experimental verification to unveil the pharmacological mechanisms underlying the therapeutic effects of LWDH pills on OP. Moreover, we aim to provide new evidence to further clarify the anti-osteoporotic mechanism of LWDH pills.

## 2. Results

### 2.1. Network Pharmacology Results

#### 2.1.1. Active Ingredients and Targets of LWDH Pills

A total of 608 active ingredients were pinpointed in LWDH pills. Subsequently, a subset of 74 active ingredients was specifically singled out, adhering to the predetermined criteria of oral bioavailability (OB) ≥ 30% and drug-likeness (DL) ≥ 0.18. Subsequently, duplicates and those without corresponding targets were excluded, leaving 42 efficient active ingredients of LWDH pills. The TCMSP database mapped these 42 active compounds individually with putative targets. The removal of duplicate targets and normalization through the Uniprot database resulted in a total of 199 targets. Furthermore, based on the number of connections (degree value) between the active compounds and targets of LWDH pills, the top 10 active compounds were identified (Table 1). These active compounds are suggested to be the core ingredients through which LWDH pills potentially exert their anti-OP effects.

#### 2.1.2. Intersecting Targets between OP and LWDH Pills

From the CTD, DisGeNET, and Genecards databases, a total of 33,134 disease targets related to OP were obtained after merging and deduplication. An intersection analysis between the targets of LWDH pills and OP identified 197 intersecting genes (Figure 1), which were considered potential biological targets for LWDH pills in the treatment of OP.

#### 2.1.3. Construction of TCM-Target Network

A total of 42 active ingredients from the six herbal ingredients of LWDH pills and 197 targets were pinpointed. An interaction network between the TCM and its targets was effectively constructed using Cytoscape 3.7.1 software (Figure 2).

#### 2.1.4. PPI Analysis and Core Targets

The 197 intersecting genes were imported to the STRING database for PPI analysis, yielding the PPI results (Figure 3A). The resultant statistical data were integrated into Cytoscape 3.7.1 software, and the top 20 hub genes were identified using the CytoHubba plugin. These include *TP53*, AKT1, *JUN*, *TNF*, *ESR1*, *IL6*, *PRKACA*, *MAPK1*, *RELA*, *EGFR*, *IL1B*, *BCL2*, *FOS*, *CTNNB1*, *CASP3*, *MYC*, *CXCL8*, *MAPK8*, *NCOA1*, and *GSK3B* (Figure 3B,C). In Figure 3B, a darker color indicates a more critical role of the gene within the network.

#### 2.1.5. Results of GO and KEGG Signaling Pathway Enrichment

An analysis of GO and KEGG was conducted on the 197 intersecting genes associated with both LWDH pills and OP. The biological function annotations of these intersecting genes associated with LWDH pills and OP are illustrated in Figure 4A–C. In terms of biological processes (BP), the genes predominantly pertained to “response to nutrient levels”, “response to oxygen levels”, “response to metal ion”, and “response to decreased oxygen levels.” For cellular components (CC), the highlighted genes were primarily related to “membrane raft”, “membrane microdomain”, and “postsynaptic membrane”. In the molecular functions (MF) domain, they primarily involved “DNA-binding transcription factor binding”, “transcription coregulator binding”, and “ligand-activated transcription factor activity”. Moreover, the KEGG pathway enrichment analysis revealed the LWDH pills-associated pathways involved in OP treatment, including the “Wnt signaling pathway”, “MAPK signaling pathway”, “MTOR signaling pathway”, and “PI3K-Akt signaling pathway” (Figure 5A). A total of 13 target genes were enriched in the Wnt signaling pathway (Figure 5B), suggesting that the “Wnt signaling pathway” could be a crucial biological mechanism responsible for the anti-OP effects of LWDH pills.

### 2.2. In Vitro Experimental Validation

#### 2.2.1. Effect of LWDH Pill-Medicated Serum on the Viability of MC3T3-E1 Cells

MC3T3-E1 cells were cultured with varying concentrations (1.25%, 2.5%, 5%, and 10%) of LWDH pill-medicated serum, and their viability was assessed at 24, 48, and 72 h using the CCK-8 assay. The results were compared with the cells grown in a standard complete culture medium, which served as a control group. The cell viability displayed no significant differences (Figure 6), indicating the LWDH pill-medicated serum did not affect the viability of MC3T3-E1 cells.

#### 2.2.2. Impact of LWDH Pill-Medicated Serum on the Levels of Osteogenic Differentiation-Related Proteins in MC3T3-E1 Cells

MC3T3-E1 cells were exposed to different concentrations of LWDH pill-medicated serum (2.5%, 5%, and 10%) and 10% standard rabbit serum (referred to as the 10% blank group). Cellular proteins were harvested after 48 h of incubation. The WB assay allowed the assessment of the expression profiles of proteins, specifically collagen Ⅰ, RUNX2, and β-actin (loading control). The results revealed that the addition of 10% rabbit serum to MC3T3-E1 cells did not yield any noticeable differences in the protein levels of collagen Ⅰ and RUNX2 when compared with the control group. Conversely, the protein levels of collagen Ⅰ and RUNX2 were elevated by LWDH pill-medicated serum at concentrations of 2.5%, 5%, and 10% in MC3T3-E1 cells when compared with the blank group (as depicted in Figure 7). These findings suggest the role of LWDH pill-medicated serum in enhancing the levels of osteogenic differentiation-related proteins in MC3T3-E1 cells.

#### 2.2.3. Effect of LWDH Pill-Medicated Serum on the Levels of Wnt/β-catenin Signaling Pathway-Related Proteins in MC3T3-E1 Cells

With the complete culture medium group as a control, MC3T3-E1 cells were treated with LWDH pill-medicated serum at varying concentrations (2.5%, 5%, and 10%), and 10% normal rabbit serum served as the blank control group. Following a 48 h duration, the extraction of cellular proteins was conducted, and the levels of Wnt3, β-catenin, and β-actin (loading control) were assessed through the utilization of a WB assay. The results revealed no significant difference in the protein levels of Wnt3 and β-catenin between the control and blank serum groups; however, LWDH pill-medicated serum at concentrations of 2.5%, 5%, and 10% enhanced the protein levels of Wnt3 and β-catenin in MC3T3-E1 cells (as illustrated in Figure 8). Hence, LWDH pills can activate the Wnt/β-catenin signaling pathway in MC3T3-E1 cells.

#### 2.2.4. Impact of the Wnt/β-catenin Signaling Pathway on Augmenting the Osteogenic Differentiation of MC3T3-E1 Cells

The viability of MC3T3-E1 cells treated with DMSO and different concentrations (0.5 μM, 1 μM, 5 μM, 10 μM, 20 μM) of ICG-001 was assessed through the CCK-8 method. The results revealed reduced cell viability in response to ICG-001 at 10 μM and 20 μM concentrations (Figure 9), suggesting that ICG-001 suppressed the proliferation of MC3T3-E1 cells. The expression of the osteogenic differentiation-related proteins and the ALP activity were subjected to further analysis after MC3T3-E1 cells were co-treated with LWDH pill-medicated serum and ICG-001.

The WB results indicated that 10 μM of ICG-001 suppressed the protein levels of collagen Ⅰ and RUNX2 and diminished the ALP activity. Moreover, ICG-001 also counteracted the LWDH pill-medicated serum-triggered upregulation in the protein levels of collagen I and RUNX2, as well as ALP activity. These findings suggest that inhibition of the Wnt/β-catenin-mediated transcription attenuated the osteogenic differentiation-promoting effects of LWDH pill-medicated serum on MC3T3-E1 cells (Figure 10 and Figure 11).

The results of ARS staining demonstrated intense positive staining of the irregular red calcified nodules in MC3T3-E1 cells treated with 5% LWDH pills (Figure 12). Additionally, these calcified nodules exhibited intense positive staining, indicating the potent osteogenic differentiation-promoting effects of LWDH pills. These results suggested that LWDH pills might enhance the osteogenic differentiation capability of MC3T3-E1 cells by activating the Wnt/β-catenin signaling pathway.

## 3. Discussion

In 2019, the mortality rate associated with osteoporotic fractures in 204 countries worldwide increased by 111.16% compared to that in 1990 [17]. With the progression of global aging, this rate may continue to rise. Therefore, effective treatment for OP holds significant implications for public health systems. In this study, we employed a comprehensive approach combining network pharmacology, bioinformatics, and experimental validation to assess the osteogenic differentiation potential of MC3T3-E1 cells induced by LWDH pills and the underlying pharmacological mechanisms. Our research identified 42 active compounds in LWDH pills, including quercetin, stigmasterol, and kaempferol. These compounds were potentially the primary ingredients responsible for the anti-osteoporotic pharmacological effects of LWDH pills. Additionally, 197 targets of LWDH pills supporting its anti-osteoporotic efficacy were identified, including *TP53*, *JUN*, *TNF*, *CTNNB1*, and *GSK3B*. Results from the KEGG enrichment analysis suggest that LWDH pills might modulate the Wnt signaling pathway, MAPK signaling pathway, and PI3K-Akt signaling pathway, thereby participating in the pathological progression of OP.

In our subsequent validation experiments, we found that LWDH pills did not affect the viability of MC3T3-E1 cells. Moreover, LWDH pills enhanced the expression of osteogenic differentiation-related proteins collagen I and RUNX2 in MC3T3-E1 cells. Additionally, LWDH pills were demonstrated to upregulate ALP activity and strengthen calcification. In terms of mechanism validation, LWDH pills were found to upregulate the expression of marker proteins Wnt3 and β-catenin in the Wnt/β-catenin signaling pathway. Therefore, we propose that LWDH pills might promote the osteogenic differentiation of MC3T3-E1 cells by upregulating the Wnt/β-catenin signaling pathway, thereby exerting anti-osteoporotic pharmacological effects.

Under physiological conditions, bone resorption and formation are in a dynamic equilibrium. The occurrence of OP may be a result of the disruption in the balance between bone resorption and bone formation, potentially resulting in a decrease in bone mass [18,19]. LWDH pills have been highlighted to promote osteogenic differentiation, which is beneficial in maintaining bone homeostasis. The Wnt/β-catenin signaling pathway is a canonical Wnt pathway, with β-catenin as a crucial factor [20]. Cell proliferative capacity, apoptotic potential, and tissue repair are regulated by the Wnt/β-catenin signaling pathway, which also augments the proliferative capacity and differentiation of bone marrow MSCs and osteoblasts [21,22]. Activation of the Wnt/β-catenin signal can stimulate osteoblast proliferative properties and facilitate the differentiation of bone marrow MSCs into osteoblasts [23,24]. Additionally, Wnts can prevent apoptotic events in mature osteoblasts [25]. Li et al. revealed that β-catenin facilitates osteogenesis and substantially enhances the ALP activity [26]. RUNX2 is a specific transcription factor for osteoblasts, and it is closely related to the proliferative capacity, differentiation, and bone formation of osteoblasts [27,28]. According to a previous study, inhibition of the Wnt/β-catenin signaling pathway can reduce the expression of RUNX2, thereby impeding the proliferative potential of osteoblasts and bone formation [29]. The binding of Wnt to its receptors, Lrp and Frizzled (FZD), results in the non-phosphorylation of β-catenin followed by its cytosolic accumulation and translocation to the nucleus [30]. β-catenin binds to the transcription factor (Tcf/Lef) and initiates the expression of target genes, including the osteogenesis-related gene *RUNX2* [30]. The expression of the osteogenic marker gene *ALP* bolsters concomitantly with the expression of the *RUNX2* gene [30,31]. *ALP* facilitates the synthesis of new collagenous matrix and the release of matrix-bound vesicles to regulate matrix mineralization by the osteoblasts [32]. Upon matrix encapsulation, the osteoblasts transform into osteocytes. Our results demonstrated that the intervention of MC3T3-E1 cells with LWDH pill-medicated serum led to an increased expression of osteogenic proteins (RUNX2 and collagen Ⅰ) and an enhancement of ALP activity. In terms of pathway validation, our cellular experiments also confirmed that LWDH pills upregulated the expression of Wnt3 and β-catenin. Consequently, we concluded that LWDH pills facilitated osteogenic differentiation in MC3T3-E1 cells by upregulating the Wnt/β-catenin signaling pathway, which might be one of the mechanisms of LWDH pills to confer their anti-OP effects. Xia et al. reported the mitigation of bone loss by LWDH pills in ovariectomized SD rats [33] and the potential relationship between the mechanism and modulation of the Wnt/β-catenin signaling pathway. Therefore, our results are consistent with these findings [33].

A few limitations of the present study should be acknowledged. First, aside from the Wnt/β-catenin signaling pathway, our research did not investigate other potential pathways (e.g., IL-17 signaling pathway and P53 signaling pathway). Second, our study utilized LWDH pill-medicated serum as the intervention agent without evaluating the active compounds within LWDH pills. Third, the lack of in vivo experiments assessing the effects of LWDH pills on OP treatment undermines the reliability of our findings. In this study, we did not detect an effect of LWDH pills on the mRNA expression level of osteogenic-related genes. In the future, we will further explore whether LWDH pills enhance the osteogenic ability of osteoblasts by increasing the transcription level of osteogenic-related genes or by inhibiting protein degradation. Thus, future studies should investigate the underlying mechanisms of LWDH pills in the treatment of OP while addressing the above limitations.

## 4. Methods

### 4.1. Network Pharmacology and Bioinformatics Analysis

#### 4.1.1. Active Ingredients and Targets of LWDH Pills

The active ingredients of each TCM were identified by utilizing the TCMSP database [34]. The appropriate active ingredients were determined based on specific screening criteria, which included a minimum OB of 30% and a DL value of at least 0.18. Based on the search functionality of the TCMSP database, the active ingredients of LWDH pills were systematically matched with their putative targets. For target normalization, the Uniprot database was employed, and the species for these targets was restricted to “Homo sapiens”. Thereby, the therapeutic targets of the active ingredients in LWDH pills were ascertained.

#### 4.1.2. Disease Targets for OP

The acquisition of OP-associated disease genes was performed by utilizing “Osteoporosis” as the search term in three databases, namely Genecards, Comparative Toxicogenomics Database (CTD), and DisGeNET. The final OP-associated gene was obtained after eliminating any identified duplicate entries.

#### 4.1.3. Intersecting Targets between LWDH Pills and OP

The resultant therapeutic and disease targets were intersected to identify the targets associated with the therapeutic effects of LWDH pills on OP. Visualization of the intersected targets was accomplished using the “VennDiagram” package.

#### 4.1.4. Visualization Network of LWDH Pills-Target Proteins

A network diagram was developed using Cytoscape 3.7.1 software by importing the ingredients and targets of LWDH pills. This diagram effectively illustrated the molecular mechanisms underlying the pharmacological effects of LWDH pills through a visual representation of the connection between the drug and its targets.

#### 4.1.5. Protein-Protein Interaction (PPI) Analysis

The genes associated with LWDH pills in treating OP were uploaded on the STRING platform (https://string-db.org/, accessed on 14 October 2023) to establish a PPI network. Key proteins with an interaction score over 0.9 were identified. The resulting PPI data obtained from the STRING database were then imported to Cytoscape 3.7.1 software for further visualization. With the Cytohubba plugin (based on the Degree algorithm), the identification of the top 20 hub genes, characterized by the highest connectivity within the PPI network, was successfully accomplished. The score increased with the intensity of the red color. The top 20 genes with the highest scores were regarded as the core target genes of LWDH pills for treating OP.

#### 4.1.6. Gene Ontology (GO) and Kyoto Encyclopedia of Genes and Genomes (KEGG) Enrichment Analyses

The GO and KEGG enrichment analyses of the intersecting genes between LWDH pills and OP were conducted using the “ClusterProfiler” package. Molecular function (MF), biological process (BP), and cellular component (CC) were included in the GO enrichment analysis. Additionally, the KEGG pathway enrichment analysis of the intersecting genes was carried out to gain a comprehensive insight into the potential biological pathways associated with the treatment of OP using LWDH pills. The statistical significance of the enrichment results was determined based on a *p*-value below 0.05. The entire enrichment analysis was executed utilizing R software (Version 4.3.1, The R Foundation; http://www.R-project.org, accessed on 3 May 2023).

### 4.2. Experimental Validation

#### 4.2.1. Experimental Animals

Three female New Zealand rabbits (aged 3 months), weighing 2.5–3.0 kg, were purchased from Huadong Xinhua Experimental Animal Breeding Farm (Guangzhou, China; License number: 44007600007678). The rabbits were housed in an SPF environment at the Laboratory Animal Center of the Guangdong Academy of Traditional Chinese Medicine (Guangzhou, China; License number: SYXK (Yue)—2019-0023). These experimental animals were utilized to extract LWDH pill-medicated serum. The experimental protocols were approved by the Animal Ethics Committee of Guangdong Provincial Hospital of Traditional Chinese Medicine (Approval number: 2018035).

#### 4.2.2. Drugs and Reagents

LWDH pills were sourced from Zhongjingyuanxi Pharmaceutical Co., Ltd. (Nanyang, China, National Medicine Permit No.: Z41022128). The CCK-8 kit was procured from Beyotime (Shanghai, China; C0038). Fetal bovine serum (10099141), 0.25% trypsin with EDTA (25200072), and penicillin-streptomycin solution (15140-122) were acquired from Gibco (New York, NY, USA). Additionally, MEM-α culture medium (iCell-0003) was purchased from iCell Bioscience (Shanghai, China), while the primary antibodies RUNX2 (12556), β-catenin (8480), and β-actin (3700S) were purchased from CST (USA). The anti-collagen I antibody was obtained from Abcam (Waltham, MA, USA; ab34712). MC3T3-E1 cells, which are mouse embryonic osteoblast precursors, were acquired from iCell Bioscience.

#### 4.2.3. Preparation of LWDH Pill-Medicated Serum

The prescribed dosage for LWDH pills was set at 4.32 g/d. Based on the dose equivalency conversion formula between humans and rabbits presented in the “Pharmacological Experimental Methodology”, a dosage of 0.24 g/kg (0.072/0.304) for LWDH pills was determined. A saline solution of LWDH pills at a concentration of 0.1 g/mL was prepared. Two rabbits were administered the LWDH Pill solution via intragastric administration once daily for a total of seven days, while the third rabbit was given an equivalent volume of saline solution. The rabbits were anesthetized with isoflurane 1 h after the last intragastric administration, and blood samples were collected from the ear artery. The samples were kept at room temperature for 2 h, and then subjected to centrifugation at 3000 rpm for 10 min. Consequently, the collected supernatant was inactivated in a water bath at 56 °C for 30 min, sterilized through a 0.22 μm filter, and stored at −80 °C for further use.

#### 4.2.4. Cell Counting Kit-8 (CCK-8) Methods

A density of 2 × 10^3^ cells/well was maintained for the seeding of MC3T3-E1 cells in a 96-well plate. Following the specified durations of drug treatment, each well received 10 μL of CCK-8 solution for a 2 h incubation. The absorbance was measured at 450 nm using a multifunctional microplate reader (Tecan, Männedorf, Switzerland) while prohibiting light exposure during the measurement process.

#### 4.2.5. Western Blot (WB) Method

Following the designated pharmacological treatments, MC3T3-E1 cells were lysed and harvested in the WB and IP cell lysis buffer (P0013, Beyotime, Shanghai, China) for protein extraction. Subsequently, the bicinchoninic acid (BCA) quantification assay (P0012, Beyotime) was employed to estimate the protein concentrations. Following that, the total protein, amounting to 20 μg, underwent SDS-PAGE for analysis. The resultant protein bands were transferred onto a PVDF membrane. In an ambient environment, the membrane was blocked using 5% skimmed milk for 1 h. Thereafter, the diluted primary antibodies, including anti-collagen I (diluted at 1:1000, ab260043, Abcam), RUNX2 (diluted at 1:1000, 12556, CST), anti-Wnt3 antibody (diluted at 1:1000, ab116222, Abcam), β-catenin (diluted at 1:1000, 8480, CST), and β-actin (diluted at 1:2000, 3700, CST; loading control) were added and incubated overnight at 4 °C. The PVDF membrane was subjected to three washes with 1 × TBST. Subsequently, another incubation at ambient temperature lasting 1 h was carried out employing horseradish peroxidase-conjugated secondary antibodies targeting either rabbit or mouse (7074, 7076, CST). Finally, the membrane and ECL (WBKLS0100, Merck, Germany) were incubated, and the chemiluminescent spectra were captured and illuminated using the imaging system (Bio-Rad, Hercules, CA, USA).

#### 4.2.6. Osteogenic Differentiation and Mineralization Assessment

A 6-well plate, designated for osteogenic induction, was coated with gelatin, with each well being seeded with 1 × 10^5^ MC3T3-E1 cells. The osteogenic differentiation of MC3T3-E1 cells, initiated by utilizing the osteogenic differentiation medium kit from Saiye Biotechnology Co., Ltd. (Suzhou, China), was commenced at a confluency of 70–80%. Serum containing varying concentrations of LWDH pill solution, either alone or in combination with ICG-001 at 10 μM, was incorporated into the osteogenic differentiation medium. Wells containing DMSO served as control. On the 30th day of differentiation, the mineralization of calcium nodules was evaluated using Alizarin Red S (ARS) solution. The images depicting mineralization were captured under an inverted microscope (ECLIPSE Ti2-E, Nikon, Tokyo, Japan). Finally, the ARS staining was eluted using a 10% cetylpyridinium chloride solution, and absorbance at 562 nm was measured using a microplate reader.

#### 4.2.7. Alkaline Phosphatase (ALP) Activity Assay

MC3T3-E1 cells were seeded in 6-well plates at a density of 1 × 10^5^ cells per well. Subsequently, after a 24 h incubation period, the regular culture medium was renewed with osteogenic differentiation medium. The medium was supplemented with 5% LWDH pill-medicated serum. Wells containing DMSO served as the control group. The ALP activity assessment on the 10th day of differentiation was performed by utilizing the ALP staining kit (Beyotime). Following that, the measurement of ALP absorbance at 405 nm was carried out using a microplate reader. The flow chart of this study is shown in Figure 13.

#### 4.2.8. Statistical Analysis

The utilization of SPSS 19.0 and GraphPad Prism 8.0 software facilitated data analysis and visualization. The means among multiple groups were compared using a one-way analysis of variance (ANOVA). To perform pairwise comparisons between the groups, the LSD-*t* test was deployed. Statistical significance in this study was determined by a *p*-value below 0.05.

## 5. Conclusions

This study highlights that LWDH pills drive the expression of osteogenic differentiation proteins RUNX2 and collagen Ⅰ in MC3T3-E1 cells. The underlying mechanisms involve the upregulation of the Wnt/β-catenin signaling pathway; however, there is a scope for future studies to further explore the active ingredients of LWDH pills and ascertain their efficacy in the treatment of OP.

## Figures and Tables

**Figure 1 pharmaceuticals-17-00099-f001:**
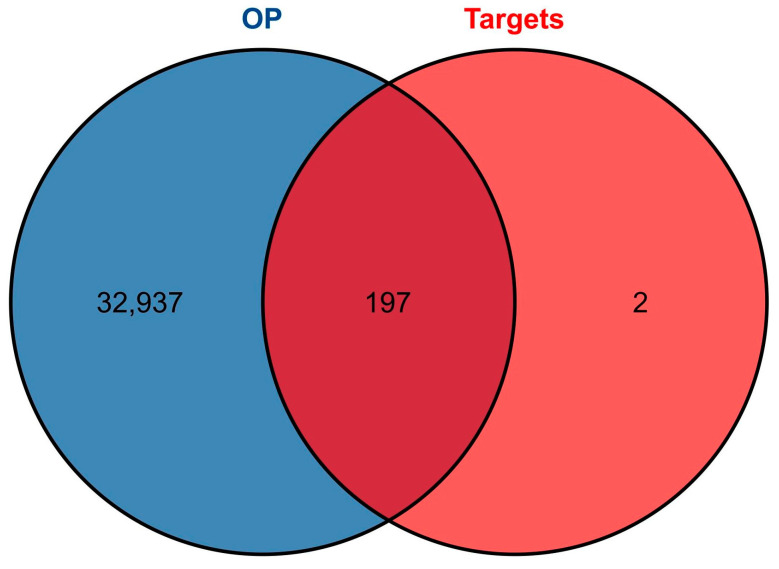
Venn diagram illustrating the intersecting targets between Liuwei Dihuang pills (LWDH pills) and osteoporosis.

**Figure 2 pharmaceuticals-17-00099-f002:**
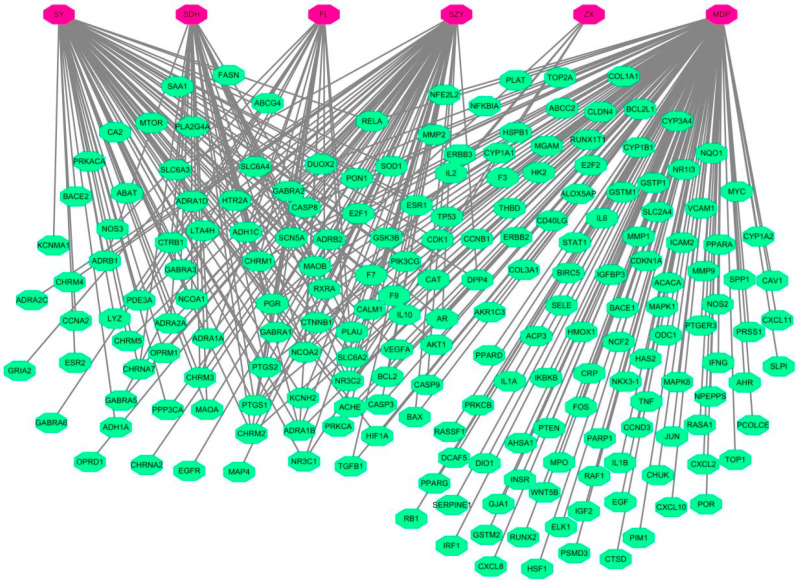
Visualization of the LWDH pills–target interaction network (herbal ingredients of LWDH pills and the associated targets are highlighted in red and green colors, respectively).

**Figure 3 pharmaceuticals-17-00099-f003:**
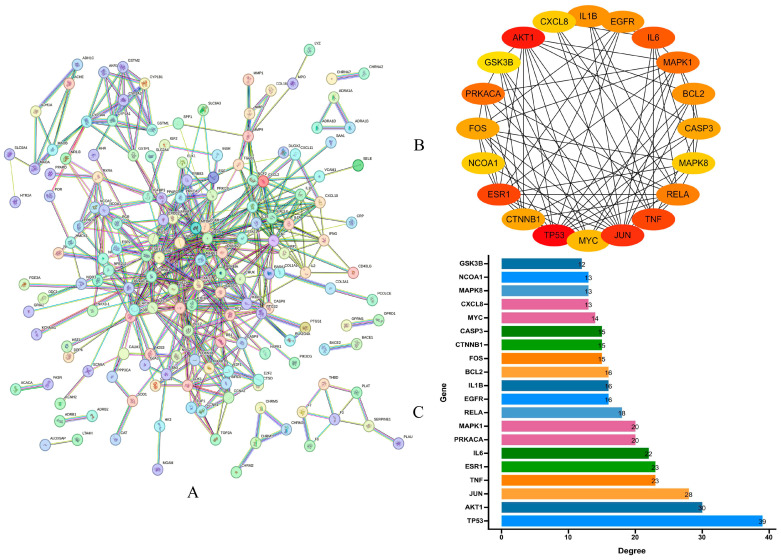
Protein–protein interaction (PPI) network analysis. (**A**) Comprehensive PPI network visualization; (**B**) list of the top 20 hub genes; (**C**) bar chart representing the significance of the top 20 hub genes.

**Figure 4 pharmaceuticals-17-00099-f004:**
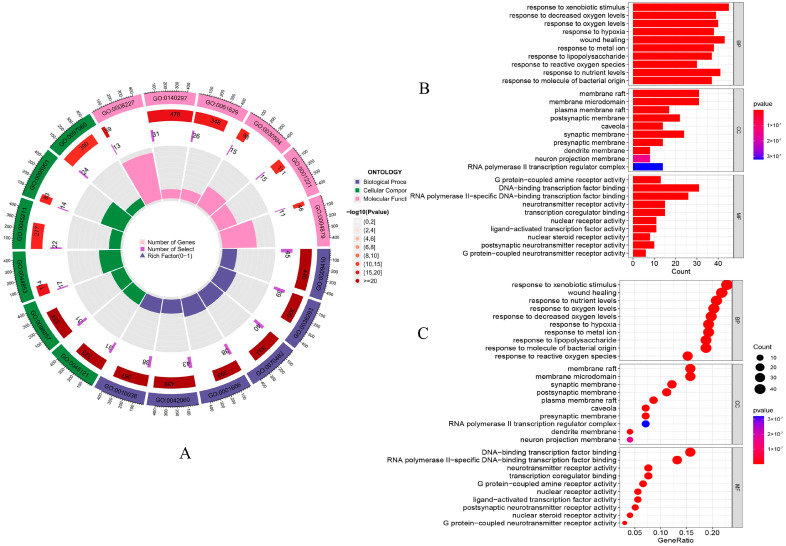
Gene ontology (GO) analysis of identified targets. (**A**) Circle bar plot displaying GO terms; (**B**) bar plot for GO analysis highlighting biological processes (BP), cellular components (CC), and molecular functions (MF); (**C**) bubble chart depicting the relative importance of various GO terms.

**Figure 5 pharmaceuticals-17-00099-f005:**
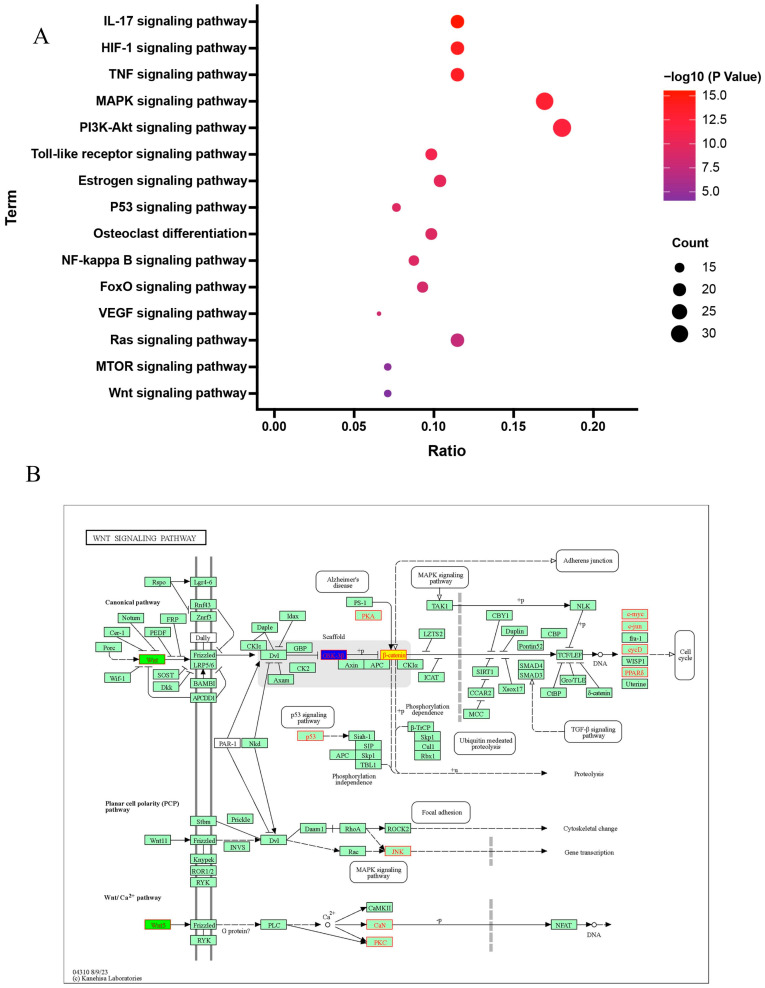
Pathway enrichment analysis using KEGG. (**A**) Overall results of KEGG enrichment; (**B**) potential targets and mechanisms of LWDH pills in the Wnt/β-catenin signaling pathway.

**Figure 6 pharmaceuticals-17-00099-f006:**
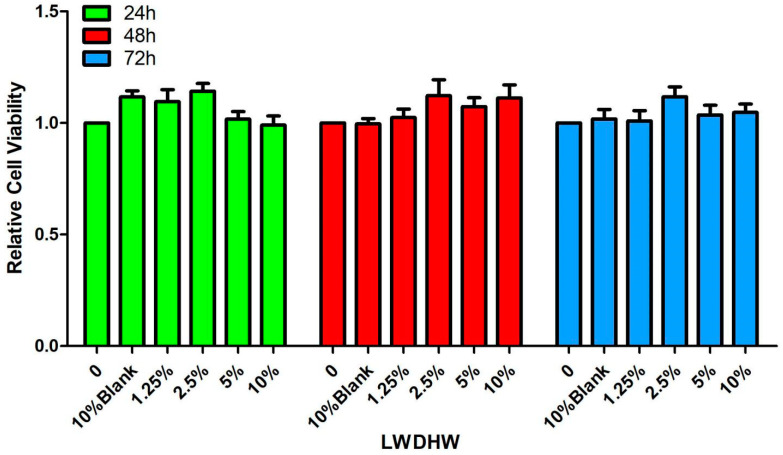
Impact of LWDH pill-medicated serum on the viability of MC3T3-E1 cells.

**Figure 7 pharmaceuticals-17-00099-f007:**
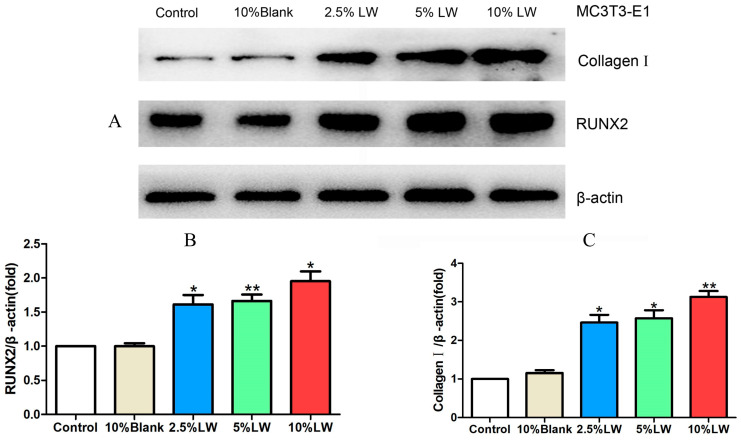
Effect of LWDH pill-medicated serum on the expression profiles of osteogenesis-associated proteins in MC3T3-E1 cells. (**A**) Representative immunoblots of different treatment groups; (**B**) quantitative analysis of protein levels of RUNX2; (**C**) quantitative analysis of protein levels of collagen I. * *p <* 0.05, ** *p <* 0.01.

**Figure 8 pharmaceuticals-17-00099-f008:**
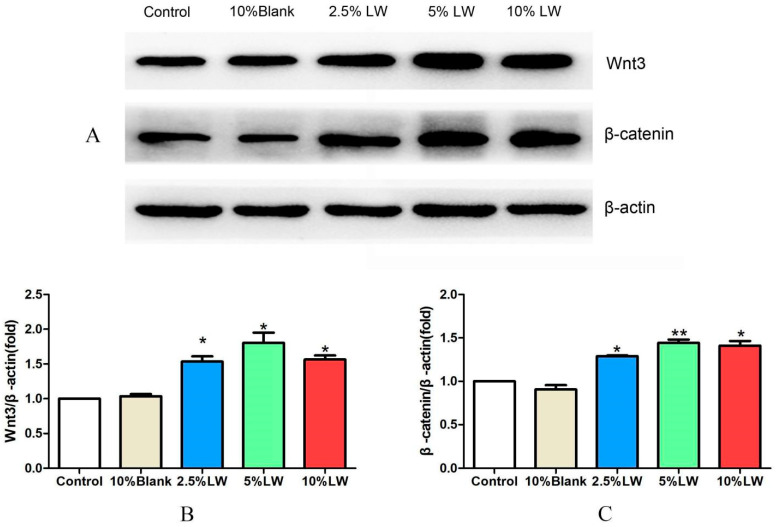
Effect of LWDH pill-medicated serum on the levels of Wnt/β-catenin signaling pathway-related proteins. (**A**) Representative immunoblots of different treatment groups; (**B**) quantitative analysis of protein levels of Wnt3; (**C**) quantitative analysis of protein levels of β-catenin. * *p <* 0.05, ** *p <* 0.01.

**Figure 9 pharmaceuticals-17-00099-f009:**
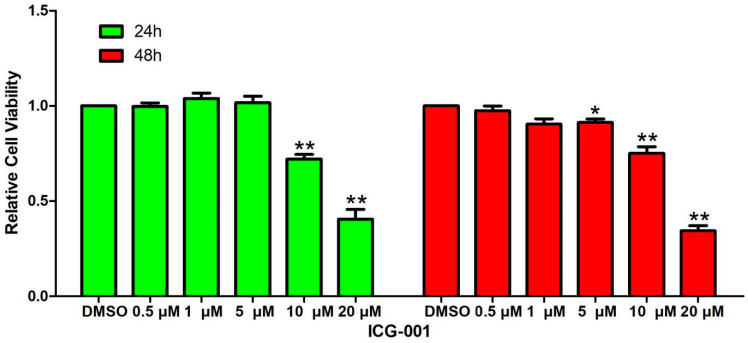
Influence of ICG-001 on the viability of MC3T3-E1 cells. * *p <* 0.05, ** *p <* 0.01.

**Figure 10 pharmaceuticals-17-00099-f010:**
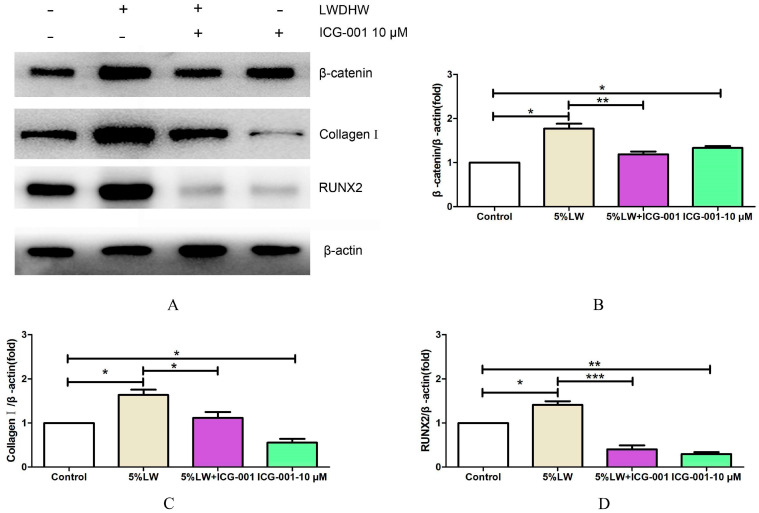
Assessment of the role of Wnt/β-catenin signaling in facilitating the osteogenic differentiation of MC3T3-E1 cells. (**A**) Representative immunoblots of different treatment groups; (**B**–**D**) quantitative analysis for the protein levels of β-catenin, collagen I, and RUNX2. * *p <* 0.05, ** *p <* 0.01, *** *p <* 0.001.

**Figure 11 pharmaceuticals-17-00099-f011:**
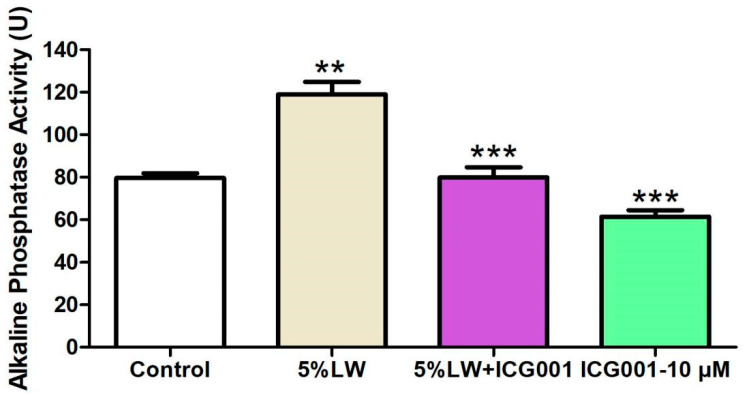
Quantitative analysis of the alkaline phosphatase (ALP) enzymatic activity in MC3T3-E1 cells. ** *p <* 0.01, *** *p <* 0.001.

**Figure 12 pharmaceuticals-17-00099-f012:**
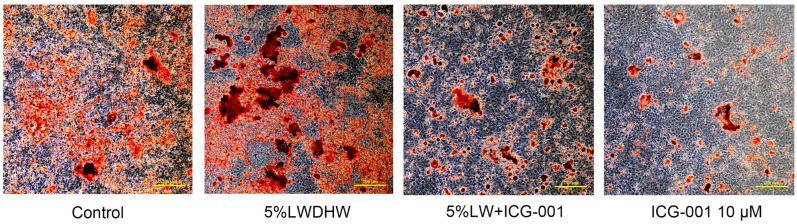
Visualization of mineralized nodule formation in MC3T3-E1 cells.

**Figure 13 pharmaceuticals-17-00099-f013:**
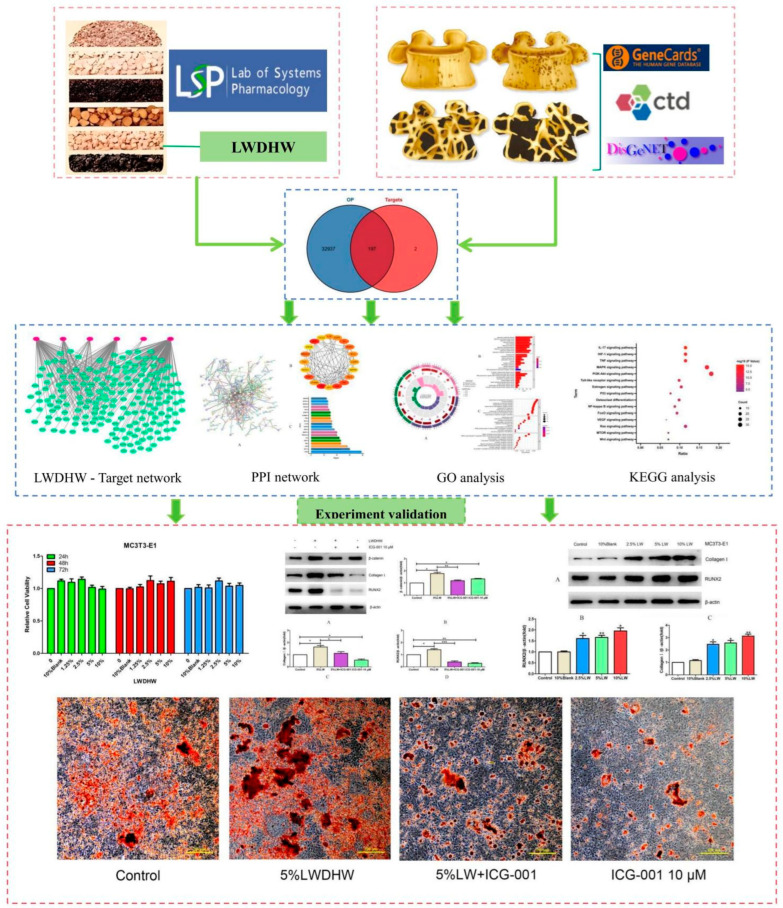
Detailed framework illuminating the mechanistic underpinnings of Liuwei Dihuang pills (LWDH pills) in osteoporosis treatment. * *p <* 0.05, ** *p <* 0.01, *** *p <* 0.001.

**Table 1 pharmaceuticals-17-00099-t001:** Ranking of the top 10 bioactive compounds based on their connectivity (degree value) within the network.

Molecule Name	Source	Structure	Molecular Formula	Molecular Weight (g/mol)	OB (%)	DL	Degree
quercetin	MDP	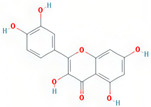	C_15_H_10_O_7_	302.23	46.43	0.28	154
stigmasterol	SDH/SY/SZY	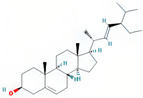	C_29_H_48_O	412.7	43.83	0.76	31
kaempferol	MDP	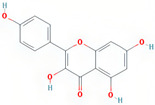	C_15_H_10_O_6_	286.25	41.88	0.24	63
beta-sitosterol	SZY	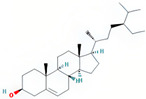	C_29_H_50_O	414.79	36.91	0.75	38
tetrahydroalstonine	SZY	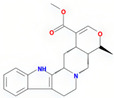	C_21_H_24_N_2_O_3_	352.47	32.42	0.81	28
kadsurenone	SY	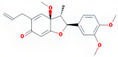	C_21_H_24_O_5_	356.45	54.72	0.38	27
hederagenin	FL	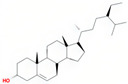	C_30_H_48_O_4_	414.79	36.91	0.75	24
hancinone C	SY	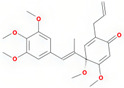	C_23_H_28_O_6_	400.51	59.05	0.39	22
diosgenin	SY	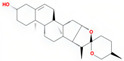	C_27_H_42_O_3_	414.69	80.88	0.81	16
AIDS180907	SY	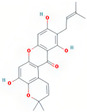	C_23_H_22_O_6_	394.45	45.33	0.77	13

## Data Availability

Data is contained within the article.

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
