# Peer review of "Liuwei Dihuang Pills Enhance Osteogenic Differentiation in MC3T3-E1 Cells through the Activation of the Wnt/β-Catenin Signaling Pathway"

_pharmaceuticals, 2024, doi:10.3390/ph17010099_

Round 1

Reviewer 1 Report

Comments and Suggestions for Authors

This study analyzes the effects of LWDH Pills and its active ingredients on osteoblast differentiation using a multifaceted evaluation method. The story of the results is smooth. Although Chinese herbal medicines have existed since ancient times and many of them have been recognized for their health benefits, functional analysis using the latest technology is still insufficient. Therefore, from this point of view, this study is very interesting in analyzing the function of LWDH Pills. 

This paper targets osteoporosis. Most of the drugs currently used to treat it target osteoclasts. 

Therefore, the discovery of compounds and the development of drugs that promote bone formation by focusing on osteoblasts are desired, but have not been very successful. From this perspective, it would be original and very meaningful to evaluate the effects of LWDH Pills and its active ingredients on the osteogenic potential of osteoblasts. 

There have been many papers that have examined the effects of Chinese herbal medicines and their components on bone cell differentiation, but this study is novel and the paper itself is well written. 

To make this paper even better, I would like to list below the areas that need to be improved. 

1. The size of the graphs and band size of western blots are large throughout this manuscript. 

2. The authors show the protein levels of Runx2, Collagen I, and Wnt3. Do the mRNA levels of these three factors also change?  

3. On the contrary in some charts such as Figure 3 and Figure 4, the text is too small to be recognized and requires improvement. 

Author Response

Dear Reviewer #1,

On behalf of my co-authors, we thank you very much for giving us an opportunity to revise our manuscript. We appreciate you very much for their positive and constructive comments and suggestions on our manuscript entitled Liuwei Dihuang Pills Enhance Osteogenic Differentiation in MC3T3-E1 Cells Through the Activation of the Wnt/β-catenin Signaling Pathway (Manuscript ID: pharmaceuticals-2732383). Those comments are all valuable and very helpful for revising and improving our paper, as well as the important guiding significance to our researches. We have studied comments carefully and have made correction which we hope meet with approval. Revised portion are marked in red in the manuscript. Appended to this letter is our point-by-point response to the comments raised by the you.

Comment 1: This study analyzes the effects of LWDH Pills and its active ingredients on osteoblast differentiation using a multifaceted evaluation method. The story of the results is smooth. Although Chinese herbal medicines have existed since ancient times and many of them have been recognized for their health benefits, functional analysis using the latest technology is still insufficient. Therefore, from this point of view, this study is very interesting in analyzing the function of LWDH Pills.

Response: Thanks for your affirmations. Your comment is a high-level summary of this manuscript, which will inspire us to further explore the clinical application value of LWDH Pills in OP treatment.

Comment 2: This paper targets osteoporosis. Most of the drugs currently used to treat it target osteoclasts. Therefore, the discovery of compounds and the development of drugs that promote bone formation by focusing on osteoblasts are desired, but have not been very successful. From this perspective, it would be original and very meaningful to evaluate the effects of LWDH Pills and its active ingredients on the osteogenic potential of osteoblasts.

Response: Thanks for your affirmations.

Comment 3: There have been many papers that have examined the effects of Chinese herbal medicines and their components on bone cell differentiation, but this study is novel and the paper itself is well written. To make this paper even better, I would like to list below the areas that need to be improved.

Response: Thanks for your affirmations. Based on your professional comments, we have made revisions to this paper.

Comment 4: The size of the graphs and band size of western blots are large throughout this manuscript.

Response: Thank you for your meticulous suggestions to improve the quality of our manuscript. Based on your comment 4 and comment 6, we have adjusted the size of the figures to ensure that all figures can be clearly recognized.

Comment 5: The authors show the protein levels of Runx2, Collagen I, and Wnt3. Do the mRNA levels of these three factors also change?

Response: Thank you for your profound questions which help improve the quality of our manuscript. Proteins, rather than DNA or mRNA, are the main substances that perform biological functions. In this study, we explored the effect of LWDH Pills on the osteogenic ability of MC3T3-E1 osteoblasts. Therefore, we focused on testing the expression of osteogenic-related proteins to demonstrate that LWDH Pills can promote the expression of osteogenic-related proteins and further enhance the osteogenic ability of osteoblasts. We will continue to investigate the effect of LWDH Pills on the mRNA expression level of osteogenesis-related genes, further investigate the mechanism of action of LWDH Pills, and explore whether these Pills enhance the osteogenic ability of osteoblasts by increasing the transcription level of osteogenesis-related genes or by inhibiting protein degradation. Your comment is very important. Therefore, we have also stated this point in the Limitations section and provided explanations for future research directions.

Revised in the manuscript (Page 15, Line 329-333): 

In this study, we did not detect the effect of LWDH Pills on the mRNA expression level of osteogenic-related genes. In the future, we will further explore whether LWDH Pills enhance the osteogenic ability of osteoblasts by increasing the transcription level of osteogenic-related genes or by inhibiting protein degradation.

Comment 6: On the contrary in some charts such as Figure 3 and Figure 4, the text is too small to be recognized and requires improvement.

Response: Thank you for your valuable suggestions to improve the quality of our manuscript. Based on your comment 6 and comment 4, we have adjusted the size of the figures to ensure that all figures can be clearly recognized.

We e would like to express our great appreciation to you for comments on our paper. And we have tried our best to revise our manuscript according to the comments. In order to meet the language requirements of English papers, we have invited American Journal Experts to modify and polish this article. Once again, thank you very much for your comments and suggestions. If this article needs further revision, please do not hesitate to contact us.

Looking forward to hearing from you. Thank you and best regards.

Yours sincerely,

Jun Liu, Lingfeng Zeng, Weiyi Yang

Reviewer 2 Report

Comments and Suggestions for Authors

This is a very interesting article, which evaluates the effect of some supplements more precisely, Liuwei Dihuang Pills used in traditional Chinese medicine on osteoporosis. This was done by evaluating the possibility to Enhance Osteogenic Differentiation in 2 MC3T3-E1 Cells Through the Activation of the Wnt/β-catenin 3 Signaling Pathway

Some comments:

-          Row 65 – “ the most significant complication of osteoporosis…” – is somehow strange since fracture are considered as the only clinical manifestation.

-          Row 75-78 some affirmations are made regarding actual treatment of osteoporosis that are not quite fair: modest efficacy, allergic reactions (no such reactions reported due to LDP?) and potential cancer risk. Please reconsider this aspect

-          Row 84-85 description of effects “nourish the yin…..” should be more carefully explained for non Chinese readers

Also, the whole premise of this article, meaning that the efficacy of LWDH Pills in the treatment has been already  established is based on reference #12 that is not quite convincing regarding the effect on postmenopausal osteoporosis in the way that effectiveness is classically evaluated in modern medicine

Author Response

Dear Reviewer #2,

On behalf of my co-authors, we thank you very much for giving us an opportunity to revise our manuscript. We appreciate you very much for their positive and constructive comments and suggestions on our manuscript entitled Liuwei Dihuang Pills Enhance Osteogenic Differentiation in MC3T3-E1 Cells Through the Activation of the Wnt/β-catenin Signaling Pathway (Manuscript ID: pharmaceuticals-2732383). Those comments are all valuable and very helpful for revising and improving our paper, as well as the important guiding significance to our researches. We have studied comments carefully and have made correction which we hope meet with approval. Revised portion are marked in red in the manuscript. Appended to this letter is our point-by-point response to the comments raised by the you.

Comment 1: This is a very interesting article, which evaluates the effect of some supplements more precisely, Liuwei Dihuang Pills used in traditional Chinese medicine on osteoporosis. This was done by evaluating the possibility to Enhance Osteogenic Differentiation in 2 MC3T3-E1 Cells Through the Activation of the Wnt/β-catenin 3 Signaling Pathway.

Response: Thanks for your affirmations.

Comment 2: Row 65 - “ the most significant complication of osteoporosis…” – is somehow strange since fracture are considered as the only clinical manifestation.

Response: Thank you for your profound questions and professional comments which help improve the quality of our manuscript. Based on your comment, we have revised this. We have deleted inaccurate description (notably high rate of disability and mortality).

Revised in the manuscript (Page 2, Line 69-71): 

The most serious complication of OP is osteoporotic fracture, which primarily occurs in the thoracolumbar vertebrae, hip, and distal radius[3, 4].

Comment 3: Row 75-78 some affirmations are made regarding actual treatment of osteoporosis that are not quite fair: modest efficacy, allergic reactions (no such reactions reported due to LDP?) and potential cancer risk. Please reconsider this aspect.

Response: Thank you for your positive comments and valuable suggestions to improve the quality of our manuscript. In order to make these descriptions more objective, we have modified them.

Revised in the manuscript (Page 2, Line 78-81): 

At present, commonly used anti-OP medications in clinical settings have various limitations or adverse reactions, including the need for long-term administration, osteonecrosis of the jaw, and gastrointestinal adverse reactions [7-9].

Comment 4: Row 84-85 description of effects “nourish the yin…..” should be more carefully explained for non Chinese readers.

Response: Thank you for your valuable suggestions to improve the quality of our manuscript. In fact, this is a characteristic theory of traditional Chinese medicine. Based on your suggestion, we have revised this.

Revised in the manuscript (Page 2, Line 86-90): 

These herbal ingredients have been extensively recognized to confer therapeutic properties in TCM that include nourishing yin, tonifying the kidneys, replenishing the essence, and benefiting the marrow; in other words, they can promote the balance and stability of the human body's internal environment and enhance the activity of osteoblasts.

Comment 5: Also, the whole premise of this article, meaning that the efficacy of LWDH Pills in the treatment has been already established is based on reference #12 that is not quite convincing regarding the effect on postmenopausal osteoporosis in the way that effectiveness is classically evaluated in modern medicine.

Response: Thank you for your positive comments and valuable suggestions to improve the quality of our manuscript. As is well known, most clinical research articles on traditional Chinese medicine are published in Chinese. Therefore, we do not have more selectivity in citing references for efficacy evaluation. But your professional comment is very valuable. Based on your comment, we have made every effort to search for published clinical evaluation articles and cite them. Finally, we cited 2 references.

Revised in the manuscript (Page 2, Line 92-94): 

The efficacy of LWDH Pills in the treatment of OP has been established based on existing evidence [11, 12].

References

[11] Ge J, Xie L, Chen J, Li S, Xu H, Lai Y, et al. Liuwei dihuang pill treats postmenopausal osteoporosis with shen (kidney) yin deficiency via janus kinase/signal transducer and activator of transcription signal pathway by up-regulating cardiotrophin-like cytokine factor 1 expression. Chin J Integr Med 2018;24(6):415-22. https://doi.org/10.1007/s11655-016-2744-2.

[12]Liu Y, Wang P, Shi X, Li H, Zhang X, Zeng S, et al. Liuwei Dihuang Decoction for primary osteoporosis: A protocol for a systematic review and meta-analysis. Medicine (Baltimore) 2019;98(16):e15282. https://doi.org/10.1097/MD.0000000000015282.

We e would like to express our great appreciation to you for comments on our paper. And we have tried our best to revise our manuscript according to the comments. In order to meet the language requirements of English papers, we have invited American Journal Experts to modify and polish this article. Once again, thank you very much for your comments and suggestions. If this article needs further revision, please do not hesitate to contact us.

Looking forward to hearing from you. Thank you and best regards.

Yours sincerely,

Jun Liu, Lingfeng Zeng, Weiyi Yang

Round 2

Reviewer 2 Report

Comments and Suggestions for Authors

Updates, responses and corrections are fine with me. Thank you for answering!